# Use of Immunostimulants in Shrimp Farming—A Bioeconomic Perspective

**DOI:** 10.3390/ani15020124

**Published:** 2025-01-07

**Authors:** Héctor Rodrigo Nolasco-Alzaga, Elizabeth Monreal-Escalante, Mariel Gullian-Klanian, Juan Antonio de Anda-Montañez, Antonio Luna-González, Fernando Aranceta, Marcelo E. Araneda-Padilla, Carlos Angulo

**Affiliations:** 1Immunology & Vaccinology Group, Centro de Investigaciones Biológicas del Noroeste (CIBNOR), Av. Instituto Politécnico Nacional 195, Playa Palo de Santa Rita Sur, La Paz 23096, Mexico; rnolasco@pg.cibnor.mx (H.R.N.-A.); jdeanda@cibnor.mx (J.A.d.A.-M.); 2CONAHCYT—Centro de Investigaciones Biológicas del Noroeste (CIBNOR), Instituto Politécnico Nacional 195, La Paz 23096, Mexico; faranceta@cibnor.mx; 3Unidad Experimental, Universidad Marista de Mérida, Periférico Nte. Tablaje Catastral 13941, Mérida 97300, Mexico; mgullian@marista.edu.mx; 4Instituto Politécnico Nacional, Centro Interdisciplinario de Investigación para el Desarrollo Integral Regional (Sinaloa), Blvd. Juan de Dios Bátiz Paredes #250, Guasave 81049, Mexico; aluna@ipn.mx; 5Benchmark Genetics Chile, Área de Gestión, Control y Análisis Santa Rosa 560 Oficina 25 B, Puerto Varas 5550200, Chile; marcelo.araneda@bmkgenetics.com

**Keywords:** survival rate increase, yield improvement, economic benefits, immunostimulant, shrimp aquaculture

## Abstract

Aquaculture is growing fast because it can produce food quickly in small spaces. However, this can lead to disease outbreaks that harm both the food and the economy. To prevent this, the shrimp industry has started using immunostimulants, which boost the immune systems of the animals and help protect them from diseases. Although these tests have been successful, it is important to also look at the economic side. A thorough analysis should combine biological, technological, and financial aspects, such as costs and profits. This way, shrimp farmers can make smart decisions about whether using immunostimulants is worth the investment. This review discusses the most important immunostimulants used in shrimp farming and suggests how to evaluate their costs and benefits.

## 1. Introduction

Shrimp aquaculture plays a crucial role in global food production and is practiced under extensive, semi-intensive, intensive, and super-intensive systems worldwide^1^, mainly differentiated by the level of management, input, and infrastructure used in each system. However, the intensification of shrimp farming has resulted in disease outbreaks that hamper production. Viral and bacterial pathogens have emerged as major causes of recent disease events on a global scale, especially in Asia and Latin America [1,2]. In the Aquatic Animal Health Code [3], the priority listed diseases for shrimp are acute hepatopancreatic necrosis disease, decapod iridescent virus 1 disease, *Enterocytozoon hepatopenaei* causing disease, infectious hypodermal and haematopoietic necrosis disease, infectious myonecrosis virus disease, necrotising hepatopancreatitis (*Hepatobacter Penaei*), Taura syndrome virus disease, white spot disease, and Yellow head disease. In this regard, the estimated annual revenue loss in India due to *Enterocytozoon hepatopenaei* and white spot syndrome virus diseases was USD 567.62 and USD 238.33 M, respectively [4], which can vary when analyzed at the farm level [5]. Similarly, net revenue losses due to acute hepatopancreatic necrosis disease ranged from USD −727.56 to 672.48 ha^−1^ [6]. In a study, the economic impact of infectious myonecrosis virus disease accounted for 29.86% mortality and USD −24,822.76 ha^−1^ [7]. Another study estimated that stocking low-level Infectious hypodermal and haematopoietic necrosis virus in shrimp augmented the farm gate value (USD 67,000 ha^−1^) with respect to the higher level one [8]. Estimated losses decades ago for Taura syndrome virus and Yellow head diseases have been around USD 0.5–2.0 billion [9]. In general, the shrimp industry encounters high losses in aquaculture due to diseases. Diseases have created economic crises, making sustainable production difficult and obligating geographic relocation in the absence of cost-effective preventive and curative treatments [1]. While antibiotics have been traditionally used to combat infectious diseases and successfully control devastating outbreaks, their use raises concerns regarding microbial resistance, residues in food products and water, and overall environmental contamination [10,11,12].

Alternative strategies that minimize or eliminate the reliance on antibiotics have gained importance in addressing these challenges. One approach involves strengthening the shrimp immune system [13]. By stimulating the immune defenses, shrimp can better combat opportunistic pathogens, thus reducing the risk of microbial resistance. Immunostimulants, both natural (such as probiotics, algae/microbial- and plant-derived compounds) and synthetic (chemical molecules), have been used in shrimp aquaculture [14,15,16]. Recombinant immunostimulants have also emerged as an alternative option. In this arena, bacteria [17], yeasts [18,19], microalgae [20], and plants [21] have been utilized to produce recombinant immunostimulants for shrimp and fish aquaculture, demonstrating results in enhancing disease resistance. However, as with any immunostimulant, production and delivery costs must be carefully considered when implementing immunostimulants at the shrimp farm level, although information is scarce on this issue [22,23].

Informed decision-making is crucial for optimizing the entire farming process. The success or failure of a shrimp farming production cycle relies on various factors, including the use of immunostimulant additives to prevent or control disease-negative impacts (Figure 1). Questions, such as biomass increase and profitability that result from the use of immunostimulants and benefit–cost ratio compared to conventional farming practices need to be addressed [24,25]. Therefore, to make sound farming strategies, decisions should always be based on statistical and mathematical evidence [26]. Hence, a substantial amount of knowledge in the field of bioeconomy applied to shrimp aquaculture has accumulated through studies examining the optimal performance of shrimp farming [27,28,29,30]. In this context, the objective of the present perspective is to provide a brief overview of the most relevant immunostimulants used in shrimp farms and offer bioeconomic elements that should be considered for the development and assessment of affordable and effective immunostimulants in shrimp aquaculture.

## 2. Global Shrimp Aquaculture Production

Crustacean aquaculture is one of the rapidly expanding industries worldwide, playing a vital role in meeting the escalating demand for marine-derived food. A staggering production record of 11,237 thousand tons (K t) (live weight) secures the position of the second most sought-after marine food, second only to finfish [2]. Among crustaceans, the whiteleg shrimp (*Litopenaeus vannamei*) is the most farmed species, accounting for a substantial production of 5812 K t (live weight). This species alone represented 51.7% of the global farmed crustacean production recorded in 2020 [2].

Most of the shrimp production (excluding China which primarily adopts an intensive production system) uses the semi-intensive production system. However, due to the increasing food demand, a noticeable trend exists toward intensification of production systems [2,31,32]. Over the past few decades, extensive research in shrimp aquaculture has focused on technical and biological factors to enhance yield based on the growth rate and survival of this species. These research efforts have also been directed towards selective breeding, quality, and composition of fatty acids and protein levels in diets, control of physicochemical water parameters, probiotics, and biofloc technology [33,34].

Under these production systems, shrimp often encounter stressors that weaken their immune system, making them susceptible to infectious disease outbreaks, which have a profound impact on the productivity and economic returns of farmed shrimp [35]. Consequently, developing innovative, economical, and ecologically viable technologies has been imperative to combat harmful diseases in aquaculture. In this sense, an alternative solution is using immunostimulants to improve defense against infectious diseases. By strengthening the immune response to combat pathogens, immunostimulants offer a valuable option to preserve the productivity and economic sustainability of farmers and companies.

## 3. Immunostimulants for Shrimp Aquaculture

Immunostimulants are substances that can be of natural or chemical origin, which function to enhance and activate both specific and nonspecific defense mechanisms within an organism’s immune system [36]. These mechanisms play a vital role in bolstering the body’s ability to combat infections and diseases [37]. Immunostimulants can be administered through different methods, including injection, immersion, and food, with the latter being the most common due to its cost-effectiveness and easy application [38,39,40]. Various commercial and experimental immunostimulants offer potential options for shrimp aquaculture [37,38,39,40,41]. As the industry continues to expand, substantial efforts are directed toward reducing production costs, with particular emphasis on utilizing intensive production systems [2,42].

A wide array of immunostimulants has been extensively studied in shrimp, with some already being applied in aquaculture practices. Apart from probiotics—known microorganisms with immunostimulatory properties—various substances, including plants, polysaccharides, and recombinant proteins, have been investigated as potential immunostimulants in shrimp [37]. Typically, these supplements are administered orally through water or via injection for up to two months to observe their desired effects [43,44]. It is important to note that any immunostimulant must be recognized, processed, and capable of inducing effector immune responses within the host to enhance disease protection. Mechanistic studies in shrimp have identified several receptors that recognize immunostimulant molecules, which trigger their subsequent processing. The desired outcome of a recognized immunostimulant is the activation of regulatory mechanisms in immune cells (hemocytes), such as transduction signals involving phosphorylation, activation of transcription factors, and nuclear translocation [45,46,47]. Once activated, the immunostimulants aim to induce effector immune responses that confer protective immunity against infectious diseases [45,46,47].

### 3.1. Probiotics

To begin addressing the importance of using immunostimulants in shrimp aquaculture, probiotics should be discussed because they are conceptualized as live microorganisms that provide health benefits to the host with immunostimulation and competitive exclusion as mechanisms underlying their effects [48,49]. Bacterial and yeast-derived species have predominantly been utilized as probiotics in shrimp aquaculture. Probiotics represent the most used biotechnology for combating infectious diseases in shrimp aquaculture, with many commercial shrimp diets incorporating them. Interestingly, there are complementary concepts or innovative applications of probiotics, such as prebiotics, synbiotics, paraprobiotics, and postbiotics [50]. For instance, prebiotics are polysaccharides considered as food sources for probiotics and natural beneficial microorganisms residing in mucosae. When prebiotics and probiotics are combined before administration, they are called synbiotics. Paraprobiotics denote non-viable probiotics, while postbiotics encompass substances derived from probiotics. Nevertheless, some authors also use the terms paraprobiotics and postbiotics to refer to any microorganism or derived substance that improves health; in this context, they specifically pertain to those with immunomodulatory effects [51]. Commercial probiotics are numerous worldwide. For instance, Weifang Yuexiang Chemical Co. (Weifang, China) has two products with a price of USD 2.00–8.00 per 1 kg, consisting of Probiotic Marine Yeast Culture Oceanic Red Yeast for Aquaculture Shrimp Fish Pond (Dosage: 400–700 gr/Ton) and Probiotics Lactobacillus Powder Animal Feed Additive Kinds of Fish Shrimp Prawn (Dosage: 5–10 g/per cubic water body in the seedling period and 150–200 g per 666 m^2^ water surface (1 m depth) in the cultivation period. Both products indicate that they enhance the immune and antioxidant functions and improve the survival rate of shrimp.

### 3.2. Polysaccharides

Several polysaccharides produced by prokaryotic and eukaryotic organisms (terrestrial and aquatic) have demonstrated direct immunostimulatory activity in shrimp. Among them, β-glucans have been widely used in shrimp aquaculture. Commercial dietary β-glucans are primarily sourced from yeast, such as *Saccharomyces cerevisiae*, and algae [52,53]. The immunostimulatory activity of β-glucans depends on various factors, including their physicochemical characteristics, dosage, administration frequency, and delivery route [54]. In addition, plants, including species from families such as *Meliaceae*, *Quillajaceae*, *Fabaceae*, *Lauraceae*, *Araliaceae*, *Apiaceae*, *Campanulaceae*, *Asteraceae*, *Rosaceae*, *Asparagaceae*, *Liliaceae*, and *Myrtaceae*, have been used as natural immunostimulants [55]. In this context, polysaccharides derived from plant fruits [56] and root extracts [57,58] alone or mixed with drugs have been evaluated in shrimps and revealed immunostimulant potential and enhanced disease resistance [59]. An example of a commercial polysaccharide is Macrogard^®^, an immunostimulant additive for animal feed, rich in β-1,3/1,6 glucans produced from the yeast *Saccharomyces cerevisiae*. The β-1,3/1,6 glucans stimulate phagocytes, enhancing their activity to ingest, digest, and destroy pathogens by secreting cytokines that strengthen the immune system. During this process, lysozyme and antimicrobial peptides are intensively produced, contributing to infection protection and reinforcing the innate defenses of the animal. The price of Macrogard^®^ varies, and the recommended dose for fish and shrimp is 1000 g per ton of feed. Similar functions are described for the product Supply Brewer Yeast for Feed Additive Rich in Immune Polysaccharides from the Tianjin Huge Roc Enterprises Co., Ltd. (Tianjin, China). The approximate price is USD 800.00–1300.00. per 5 Tons, and the dosage is 40–80 kg/Ton feed.

### 3.3. Medicinal Plants

Plants possessing medicinal properties for humans are often cost-effective and have been selected for aquaculture due to their bioactive immunostimulant compounds [60]. Various parts of plants, including seeds, roots, flowers, fruits, and leaves, have been administered to fish through oral, immersion, and injection routes [61]. The dosage of plant-based/plant-derived immunostimulants plays a crucial role in immunomodulation and disease resistance in shrimp [62,63]; algae have demonstrated immunostimulant properties in several shrimp species [64]. Triple Cure-Vibriosis (USD 4.94 per kg) from Prions Biotech is an herbal medicine indicated for shrimp. It is indicated in all types of hatcheries and shrimp ponds to optimize and maintain immune competence, kill pathogens (best for vibriosis disease), and enhance survival. However, it seems that commercial products based on medicinal plants are formulated with other compounds and used as sanitizers as well.

### 3.4. Recombinant Proteins

Recombinant proteins have emerged as a promising alternative for shrimp immunostimulation [65]. Genetic engineering techniques have been utilized to produce antigens specific to viral and bacterial pathogens affecting shrimp; thus, bacteria, yeast, and microalgae are used as expression hosts [66]. Feeding animals with recombinant antigens deriving from these microbial sources has demonstrated remarkable benefits, including enhanced immune parameters and resistance to infections. For instance, the administration of recombinant VP28 of the white spot syndrome virus (WSSV) combined with poly I:C as an adjuvant increased the survival rate of kuruma shrimp (*Marsupenaeus japonicus*) [67]. Moreover, natural proteins—involved in the immune response when administered as recombinant proteins—have also shown promising results in strengthening the immune system in shrimp. An example is the effect of recombinant lysozyme protein, which increases immunological parameters when injected into shrimp, including total hemocyte count, phagocytic and respiratory burst activities, phenoloxidase activity, and lysozyme activity [68]. These findings collectively indicate the significant potential of recombinant proteins as effective immunostimulants in shrimp aquaculture. By harnessing the power of genetic engineering, these innovative strategies hold promise for advancing disease management and improving the overall health and productivity of shrimp farming practices. Despite its potential, no commercial recombinant proteins seem available for aquaculture so far.

### 3.5. Bioflocs

Bioflocs technology (BFT) in shrimp production has emerged as a sustainable practice with the potential to mitigate environmental impacts and prevent pathogen introduction. The microbial community associated with BFT plays a crucial role in nutrient detoxification, improving feed utilization, and promoting animal growth. The biofloc system typically hosts a diverse array of bacteria with cell walls comprising components such as bacterial lipopolysaccharide, peptidoglycan, and β-1, 3-glucans known to stimulate nonspecific immune activity in shrimp [69]. In addition, the utilization of different carbon sources (C1, C2, C3, C4, C5, C6, C7, C8) in the culture media has been shown to influence the synergistic effects of the factors mentioned above on improving water quality within the shrimp environment. As a result, specific immune parameters, such as total hemocyte count and prophenoloxidase (ProPo) activity, are augmented in cultured shrimp [70]. Furthermore, glucose combined with bacterial biofloc media belonging to the family Rhodobacteraceae has exerted immunostimulant effects on shrimp [71]. This combination regulates the microbiota in the shrimp gut, playing a crucial role in maintaining a healthy immune system. Including glucose in the diet, the biofloc Rhodobacteraceae media can potentially enhance immune responses in shrimp and contribute to disease prevention.

Despite the considerable research on immunostimulants in shrimp aquaculture, there is a noteworthy shortage of studies focusing on the bioeconomic analysis of their implementation. While the industry has already incorporated immunostimulants into commercial shrimp diets, the economic aspects associated with their use remain largely unexplored. Comprehensive bioeconomic analyses are imperative to evaluate the cost-effectiveness and potential benefits in shrimp farming of implementing immunostimulants.

## 4. Bioeconomy in Shrimp Aquaculture

### 4.1. Basic Concepts in Bioeconomy

The concept of bioeconomy encapsulates a dynamic economic framework characterized by the intricate interplay between the burgeoning needs of humanity and the sustainable wealth offered by the abundant natural resources [72]. Specifically, the economy is defined as where the foundational components for materials, chemicals, and energy are sourced from renewable biological resources, promoting ecological balance and resilience [73].

Considering that animal production systems rely on living organisms acquired during their early developmental stages, comprehending the biological and economic dynamics of the system assumes paramount importance [74]. Such understanding becomes a pivotal asset for informed decision-making in the management of farming enterprises. Hence, the conceptual framework of bioeconomy extends its implications to the foundational constituents of economic discourse within business management, encompassing *costs*, *revenue*, and *profit* [75]. Additionally, this framework interfaces seamlessly with the core elements of biological considerations in business management, notably *growth*, *survival*, and *biomass* [76].

### 4.2. Revenue

Revenue can be defined as the monetary proceeds derived from the commercialized yields in the production system [77]. Within aquaculture, it commonly corresponds to the organisms reaped from aquatic environments (total biomass), encompassing diverse species such as fish, crustaceans, mollusks, and other ecologically important entities. Revenue quantification is a pivotal metric in appraising the financial dynamics of bioeconomic systems, accentuating the tangible outcomes of sustainable resource utilization strategies [77,78].

### 4.3. Costs

Within the intricate tapestry of functional enterprise, the notion of costs transcends mere monetary exchanges and ventures into the multifaceted domain of resource allocation. In essence, costs signify the finances needed to procure, acquire, or uphold assets and activities pivotal to the harmonious progression of operations [79,80]. In the context of a thriving business ecosystem underpinned by the principles of bioeconomy, costs are discerned as the judicious disbursements of financial resources aimed at sustaining the dynamic life cycle of production [77,80].

### 4.4. Profit (Net Revenue)

Profit is the financial surplus obtained by deducting total costs from total revenue, serving as a measurable indicator of economic gain within the framework of bioeconomic operations. In its simplistic and harmonious form, it is mathematically expressed as *Profit_t_* = *ΣRevenue_t_* − *Σcosts_t_*, where the profit obtained at t time will be determined by all the money earned minus all the money spent at that time [77]. This quantitative differential encapsulates not only the economic feasibility of endeavors but also stands as a sentinel of efficacy, reflecting the harmony between revenue generation and the astute management of resources within the dynamic tapestry of the bioeconomy.

### 4.5. Growth

Growth is a fundamental biological phenomenon characterized by the dynamic generation and meticulous reorganization of tissues within a living organism, achieved through intricate processes of cell division and subsequent cell differentiation [81,82]. This intricate progression transpires across distinct developmental stages, primarily by genetic instructions and environmental stimulus [81,82]. In aquaculture, growth assumes an essential role as an indicator denoting the attainment of a commercially viable stage by an organism [83]. In other words, growth indicates if the organism has reached the optimal configuration of length, shape, and/or weight suitable for trading, often referred to as the “market size”.

### 4.6. Survival

In the temporal context of an initial population, be it within a natural or artificial ecosystem, survival embodies an important biological variable—it quantifies the count of organisms that endure from an initial temporal point “*a*” to a subsequent point “*b*’, juxtaposed against the original population count at time “*a*” [84]. In the domain of aquaculture, this metric assumes heightened impact as it provides a tangible gauge of the well-being continuum experienced by the cultivated organisms throughout their developmental trajectory. Hence, survival diligently mirrors the degree of efficacy in the management and care protocols utilized in the cultivation process [85].

### 4.7. Biomass

Biomass, a foundational concept in farming yield assessment, encapsulates the cumulative mass encompassed by the entirety of organisms [86]; in practice (within the context of aquaculture), this parameter assumes a tangible form by computing the product of the mean weight characterizing the cultured organisms and the extant population inhabiting the specific culture stage. Mathematically, it can be expressed as *b_t_
*= *w_t_ * n_t_*, where *w_t_* is the product of the individual mean weight at time *t*, whereas *n_t_* is the number of surviving organisms at that time [87]. In essence, biomass reflects a dynamic interplay of growth and survival intrinsic to the managed ecosystem.

To ensure the development of sustainable and economically viable shrimp aquaculture practices, considering other economic elements is also recommended, such as marginal costs and revenue, market demand, and profitability. Furthermore, incorporating ecosystem dimensions into the deliberation is imperative, addressing resource renewability and implementing effective waste management strategies. Conducting initial research on immunostimulants with these factors allows for simulating different scenarios and aids in anticipating the potential success of their implementation at the farm level. This approach provides valuable insights for decision-making, empowering stakeholders to make informed choices that optimize performance and economic outcomes.

## 5. Bioeconomic Assessments in Shrimp Aquaculture: Building the Path Towards Immunity Enhancement for Sustainable Resource Management

Considering that the fundamental goal of aquaculture farms is to achieve optimal profitability through increasing growth and survival rates and subsequent biomass of cultivated organisms, as well as the simultaneous reduction of operational expenses by minimizing energy consumption and procuring cost-effective and dependable resources, a comprehensive bioeconomic assessment becomes imperative. The execution of this bioeconomic evaluation can be generally achieved through utilizing rigorous modeling techniques and robust statistical analyses. This evaluation can be based on the type of system in which the organisms are reared, whether in a semi-intensive, intensive, or super-intensive system. For instance, Moreno-Figueroa et al. [87] demonstrated the success of an intensive farm system through a bioeconomic approach by combining a photo-heterotrophic saline system with minimal seawater replacement. Implementing a bioeconomic model, these authors estimated net revenues for two cycles per year (spring–summer and summer–autumn). Their results revealed substantial differences in net revenues and cost–benefit ratios between the two cycles. The spring–summer period yielded a profit of USD 12,600/ha and a benefit–cost ratio of 1.31, while the summer–autumn period resulted in a profit of USD 38,930/ha and a benefit–cost ratio of 1.93. These variations can be attributed to the intense sunlight during July and August—the hottest months of the year—leading to higher rates of transformity (solar energy needed to produce a specific input). Overall, productivity and economic performance accounting for the net revenue were mainly influenced by the shrimp price, followed by the final weight of shrimp, dissolved oxygen, and temperature.

Remarkably, in super-intensive systems, the use of specific immunostimulants becomes inevitable. Lima-Viera et al. [88] evaluated the implementation of probiotics with six environmental and economic indicators, including renewability, emergy investment ratio (EIR), emergy yield ratio (EYR), environmental load (EL), emergy sustainability index (ESI), and transformity for three types of shrimp intensification: semi-intensive, super-intensive, and a third type. The study found that the super-intensive system expressed the best renewability, ELR, and ESI, with values of 14.79%, 5.76, and 0.20, respectively. The superior performance per hectare in the super-intensive system was mainly attributed to the utilization of biofloc and probiotics compared to the other systems [88]. Biofloc and probiotics, which consist of beneficial microorganisms, such as Pseudomonas, Acinetobacter, Cellulomonas, Rhodopseudomonas, Nitrosomonas, and Nitrobacter, act as natural remedial agents for organic waste and are ideal for mitigating the effects of heavily loaded ponds in super-intensive systems [89]. These microorganisms effectively mitigate the accumulation of nitrites and nitrates while they also address the oxygen demand in saturated ponds, acting as shock absorbers against the detrimental effects of high stocking densities above 50 individuals/m^2^. Although this bioeconomic study did not consider immune defense parameters, probiotics are microorganisms that can strengthen the shrimp’s immune system and disease resistance. Thus, immunological variables could be considered to make the bioeconomic assessment robust.

Upon this evidence, assuming that when whatever immunostimulant enhancing survival, growth rate, or both is implemented into the system, the outcome might result in greater profits than those obtained under conventional production factors. Thus, the incorporation of novel biotechnology into farming systems virtually tends to represent a benefit for aquaculture farms. In terms of income increase and a profitable internal rate of return, certain applied biotechnological applications might improve overall finances when proven experimentally successful, which in aquaculture is usually focused on the survival rate and/or growth rate improvement of the farmed species. However, in practice, this assumption is not always true. In a marginal analysis study of larval shrimp production, Peñalosa-Martinell et al. [25] showed that although the use of homemade probiotics was accompanied by a reduction in unit production costs relative to the use of commercial probiotics, the economic benefits obtained with their use (either self-acquired or bought) were equal to those obtained without them.

On the other hand, exploring substituting a portion of the administered food with an immunostimulant additive or formulated feed holds the potential for promising results. For example, a study reported [90] that by replacing 75% of dietary live algae with a formulated feed called MySpatTM, the green-lipped mussel (*Perna canaliculus*) achieved the same level of productivity. Remarkably, this dietary substitution significantly reduced feed costs, decreasing from USD 221 kg^−1^ to 138 kg^−1^. Such findings demonstrated the significance of investigating alternative feeding strategies and formulations that can maintain or improve productivity and optimize aquaculture practices with economic efficiency. Hence, there are immunostimulants whose production costs could be reduced due to the process of manufacture. One example is recombinant immunostimulants produced in transgenic plants using the absorption of solar energy, which is free and always available [91] (Figure 2). Combining this biotechnology with oral administration of the immunostimulant (food mixed with the freeze-dried transgenic plant) could prove financially beneficial for the farmer.

## 6. Integration of Immunostimulation Technologies in Shrimp Farming Through Bioeconomic Analysis

In the bioeconomical framework throughout the preceding two decades, multiple assessments have been undertaken to ascertain the optimal outcomes for specific cases of shrimp cultivation. These assessments span diverse farming strategies, encompassing a range of production factors and innovative technologies, in addition to modeling innovation/adjustment and complementary risk analysis when uncertainty over the production outputs exists (Table 1). Most of these research endeavors have primarily concentrated on stocking density, optimal feeding, and water quality management. In contrast, the economic benefits gained by biological performance enhancement and health condition improvement through immunostimulation have received relatively nascent interest [25,92]; only recently has the sphere of comprehensive investigation crossed the threshold sustained by a bioeconomic vision.

When it comes to immunostimulation technologies, initiating the discourse on the requisite methodologies and procedures for a robust bioeconomic assessment, the starting point involves elucidating the origin and nature of the immunostimulant (for more information see section “Immunostimulants for shrimp aquaculture”). Whether synthesized in a laboratory setting or procured through commercial channels, this immunostimulant must have undergone prior assessments to determine optimal dosages and administration frequencies. Ideally, these assessments would have involved trials with shrimp populations reared under laboratory conditions and, where feasible, within environments mirroring commercial production settings to ensure biosecurity and effectiveness [103,104].

Upon the commencement and advancement of the culture cycle—marked by the initiation of growth, survival monitoring, and subsequent biomass accumulation—alongside the meticulous documentation of incurred costs and potential revenue projection upon attainment of the shrimp minimum market size, formulating and calibrating comprehensive economic and biological sub-models become essential with the structured data [105]. Thus, the combined modeling endeavor serves as the cornerstone for attaining a holistic bioeconomic perspective of the system’s productive units (bioeconomic model). An avenue to simulations of different scenarios is provided whereby the effective units could experience change or uncertainty [29].

Due to the mathematical complexity that modeling might have—as an alternative to constructing models from scratch—an equally viable approach involves selecting pre-existing models from the relevant literature (as shown in Table 1). These chosen models can then be tailored, if necessary, to align with the specific nature of the data provided and the intricacies of the production scheme under evaluation. An extension of the bioeconomic model could encompass a technological submodel, offering an enhanced analytical framework. For instance, in scenarios where the immunostimulant is blended with the administered feed, an integrated feeding sub-model analogous to the approach proposed by Seijo [29] or Esmaeili [93] could potentially yield a more comprehensive insight into operational costs, delineating the monetary inputs invested in feeding processes. Alternatively, the integration of an energy sub-model becomes indispensable in cases where immunostimulants contribute to energy conservation, such as the utilization of probiotics or biofloc to curtail energy consumption through diminished water exchange. This sub-model, as the one proposed by Peñalosa-Martinell et al. [25], effectively quantifies the energy expended throughout the production cycle, offering a nuanced evaluation of energy dynamics within the system.

As the final task with the organization of output data and successful calibration of models to accurately mirror the dynamics of production units, the subsequent phase involves delving into risk and statistical analyses applied to production outputs. Considering that certain immunostimulants display efficacy in the early shrimp growth stages but may lose effectiveness with prolonged use, dose, and frequency [106,107], a risk analysis is suggested for both scenarios: control and an immunostimulation treatment. This analysis should evaluate the potential for increased biomass and revenue by extending the culture period beyond the targeted market size. To address this issue, the Monte Carlo analysis—a computational method utilizing random sampling and statistical modeling to simulate a system or process (in this case, bioeconomic model) multiple times and generate probability distributions—emerges as a potent tool for assessing the probability of achieving optimal production (maximum yield) relative to reference limit points [29].

In this sense, in the context of post-larval grow-out within an intensive culture, a risk analysis can be conducted using a triangular distribution approach to assess the likelihood of maintaining or surpassing survival rates by the end of the production cycle. This procedure entails utilizing minimum, average, and maximum survival rate values, typically drawn from the literature or prior experiments, to estimate the probability of survival enhancement within the intensive culture. For example, if the survival rates for intensive systems spanned 70–90% [6,96,99], the triangular distribution within the Monte Carlo analysis would incorporate 70% as the minimum, 80% as the average, and 90% as the maximum value. The potential for extending the cycle duration to augment harvest biomass can be explored if the immunostimulant ensures a 90% survival rate by the cycle conclusion. Utilizing the biological submodels and triangular distribution in a Monte Carlo analysis, this approach provides probabilistic insights into the feasibility of maintaining a 90% survival rate for an extended time.

This imperative step is undertaken to culminate in a comprehensive bioeconomic analysis, encapsulating not only the intricate interplay of economic and biological factors but also the inherent uncertainties embedded within the production process (Figure 3).

## 7. Conclusions

The existing evidence strongly supports and advocates using immunostimulants in shrimp aquaculture, particularly considering incorporating orally delivered immunostimulants as feed additives. These immunostimulants have shown promising results in enhancing shrimp’s immune response and disease resistance, ultimately improving their overall health and productivity. However, the existing knowledge and going beyond are essential to reinforce these findings with robust statistical support.

To achieve this goal, a bioeconomic assessment focusing on the economic feasibility of immunostimulants in shrimp aquaculture should be considered in the foreseeable future. Such assessment should provide valuable insights into the cost-effectiveness and profitability of incorporating immunostimulants into commercial shrimp diets. By conducting a thorough analysis that encompasses both the biological and economic aspects, stakeholders can make informed decisions regarding implementing immunostimulants in shrimp farming practices.

Moreover, integrating bioeconomic elements into the evaluation of immunostimulant strategies should help simulate various scenarios and anticipate the potential success of their implementation at the farm level. This proactive approach may contribute to the sustainable growth of the shrimp aquaculture industry by ensuring optimal resource allocation, mitigating risks, and maximizing economic returns.

The optimal approach to facilitate the technology transfer of biological enhancers, such as immunostimulants, involves presenting outcomes that extend beyond mere survival enhancements. Thus, emphasizing the economic gains achieved through these interventions becomes pivotal. Demonstrating quantifiable monetary benefits to aquaculture companies can bolster their confidence in strategic investments. This emphasis on financial viability adds a critical layer to the decision-making process, fostering a favorable environment for adopting innovative solutions in the industry. Finally, through these efforts, the industry can continue to advance and thrive while maintaining a sustainable and economically viable environment to sustain shrimp aquaculture.

## Figures and Tables

**Figure 1 animals-15-00124-f001:**
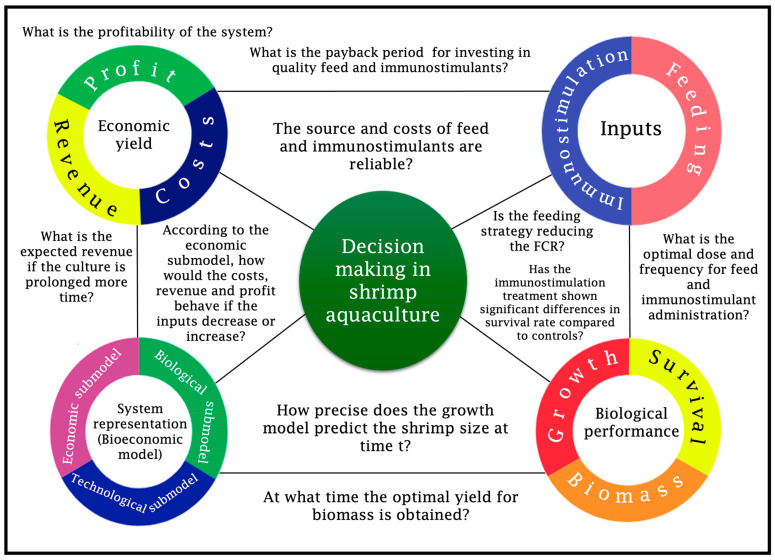
Decision-making in shrimp aquaculture. A primary concern in sustaining shrimp aquaculture systems is procuring essential inputs, primarily feed. In intensive or super-intensive systems, introducing biological enhancers like immunostimulants is crucial—they must undergo prior testing to determine the optimal dosages and frequencies and statistically establish their capacity to improve survival rates compared to control treatments. Immunostimulation effectiveness is expressed in the biological performance, encompassing growth and survival rates, collectively contributing to biomass accumulation. Ensuring the optimal dosage and feed frequency is imperative for sustaining the system efficiently, always seeking to reduce and/or keep the feed conversion ratio (FCR) of cultured shrimps. To comprehend the system dynamics, a bioeconomic model is utilized, serving as a tool to reflect and predict the system behavior concerning both biological and economic yields. With a precise representation of the production system, exploring novel approaches involving inputs and production factors becomes feasible. This exploration leads to anticipated revenues, costs, and profit calculations within the bioeconomic model.

**Figure 2 animals-15-00124-f002:**
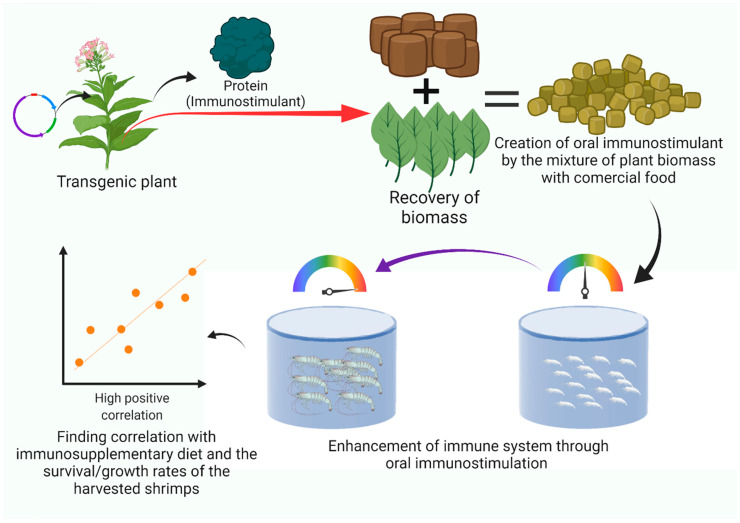
Representative example of a workflow through plant-made recombinant immunostimulant production, delivery, and immunological evaluation in shrimp culture. Initiated by the transcription of the recombinant gene, this process culminates in integrating the specific recombinant protein within the cellular machinery of the transgenic plant—tobacco, in this instance. The meticulous care and nurturing of the plant are fundamental, ultimately leading to biomass accumulation as the plant achieves the desired size. Subsequently, this biomass undergoes a transformation, coalescing with commercial food to yield pellets purposefully designed for systemic administration. As the ensuing weeks progress, a discernible correlation emerges among shrimp size, survival rate, and the efficacy of the recombinant immunostimulant-enhanced food, revealing the intricate interplay between this innovative approach and the biological outcomes within the system.

**Figure 3 animals-15-00124-f003:**
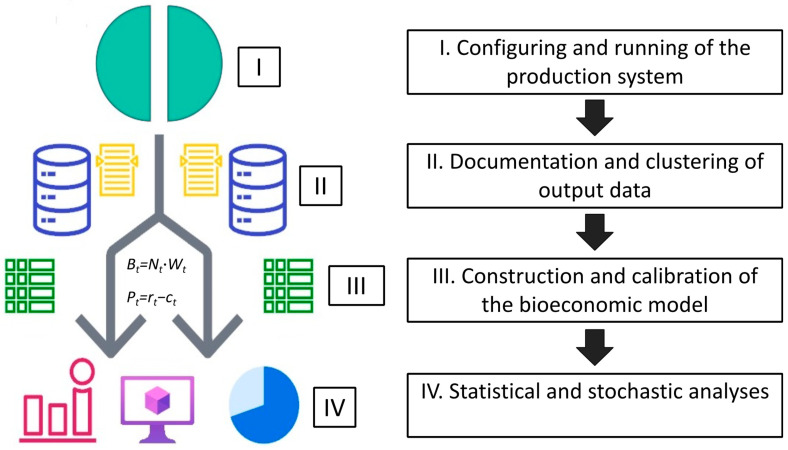
Bioeconomic assessment of an immunostimulation technology in shrimp farming. I. To initiate yield trials for both the control and immunostimulation treatments, the farm infrastructure must be configured to enable the controlled operation of two distinct segments for shrimp rearing. II. Once both systems are operational, meticulous documentation and categorization of output data should start, encompassing critical biological and economic variables, notably shrimp weight and survival, as well as fixed and variable costs. III. With the output data meticulously organized within spreadsheets, the subsequent steps involve the construction and calibration of the bioeconomic model, aiming to achieve the most optimal representation for both systems. IV. To facilitate comparison, profitability, and benefit–cost ratios should be utilized, complemented by rigorous statistical analyses of key output variables, such as shrimp weight and length. Furthermore, stochastic analysis needs to be implemented to explore the economic feasibility of extending the culture period. This figure provides a simplified overview of the comprehensive bioeconomic assessment process for evaluating the immunostimulation technology’s impact on shrimp farming.

**Table 1 animals-15-00124-t001:** Most relevant bioeconomic studies per year in shrimp farming over the past two decades. Different approaches in farming strategies, technologies, and production factors.

Study and Model Building Descriptions	Highlights	Ref.
Estimation of risk of exceeding bioeconomic limit reference points in shrimp aquaculture systems. A stocking density of 75 postlarvae/m^2^. **Biological model (survival):** $N_{t+dt}=N_{o}\;*\;e^{\left(-M*t\right)}$, where $N_{t+dt}$ is the number of surviving individuals, M is the natural mortality rate, and $N_{o}$ is the initial number of individuals. **Biological model (biomass):** $B_{t}=N_{t}\;*\;W_{t}$, where $N_{t}$ is the number of surviving individuals and $W_{t}$ is the average weight of individuals at different ages t. **Biological model (Length):** $L_{t}=L_{\infty }\;*\;(1-e^{\left(-k*t\right)})$, where k is the curvature parameter of the growth function and $L_{\infty }$ is the maximum length of species. **Biological model (weight):** $W_{t}=\alpha \;*\;L_{t}$***^β^***, where α and β are parameters of length–weight function. **Biological model (feeding):** $F_{t}=\left(B_{t}+N_{t}\left(W_{t+1}-W_{t}\right)\right)\;*\;FCR$, where FCR is the food conversion ratio and $\left(W_{t+1}-W_{t}\right)$ represents the changes in weight of individuals in the following time interval. **Economic model (Profit):** $\pi_{t}=N_{t}\;*\;p_{t}\;*\;W_{t}-\sum_{t=o}^{t}\left(cf\;*\;F_{t}+{OVC}_{t}+{FC}_{t}\right)-\left(hc+hoc\right)\;*\;N_{t}$, where $p_{t}$, cf, $F_{t}$, ${OVC}_{t},$ and ${FC}_{t}$ are the size price (USD per kg), the unit cost of food (USD per kg), harvest cost in time t, other variable costs in time t, and the fixed costs in time t, respectively. Additionally, hc and hoc are the unit cost of harvest and the unit cost of head removal, respectively. **Economic model (present value of profit):** ${PV}_{\pi_{t}}=\frac{\pi_{t}}{{\left(1+d\right)}^{t}}$, where d is the discount rate.	In this study, the inherent uncertainties contributing to the variability within estimated bioeconomic variables and parameters are seamlessly integrated by means of a Monte Carlo analysis. This numerical approach was proposed as a tool for gauging the probability of surpassing limit reference points, which in turn serve as key indicators illuminating the performance landscape within the realm of aquaculture. Utility: A simple classification of indicators and reference points is provided for decision-making process in aquaculture production systems.	[29]
Determination of the optimal feeding level for shrimp culture through a bioeconomic study. A semi-intensive system. **Economic model (Profit):** $\pi =p\left(w\right)\;*\;w\left(T\right)e^{-rT}-\int_{.}^{T}Pf\;*\;f(t)e^{-rt}dt$, In this equation, π is the present value of the profit from each shrimp, defined as revenue less costs. Revenue per shrimp is computed as $p\left(w\right)*w\left(T\right)e^{-rT}$, where $p\left(w\right)$ is the selling price, $w\left(T\right)$ is the weight at harvest time T, and r is the discount rate (i.e., the sum of instantaneous mortality and the interest rate). Costs are computed by integrating feed costs over the period of culture, that is, $Pf*f(t)e^{-rt}$, where Pf is the unit cost of feed and f(t) is the amount of feed given at time t. Feeding varies through time according to shrimp weight.	An economic comparison between the two feeding paths (conventional vs. optimal path created with the feeding model) shows that the profits of shrimp culture are 35% higher for the optimal path. Utility: Determination of the optimal feeding level for shrimp culture.	[93]
Determination of an optimal harvesting strategy to maximize annual net revenue from super-intensive recirculating shrimp production systems. Stocking density of 301–705 postlarvae/m^2^. **Biological model (Survival model, shrimps/m^2^):** $Q_{i}=v_{i}Q_{i-1}$, where $v_{i}$ was the weekly survival rate in % and $Q_{i}$ is the number of shrimps at the end of week one and, thus, also at the beginning of week two, etc. **Economic model (Revenue):** $R_{i}=P_{j}{\;*\;W}_{i}\;*\;H_{i}$, where $P_{j}$ is the price of different shrimp size categories, $W_{i}$ is the weight of the shrimp at the end of the week i, and $H_{i}$ is the decision variable denoting the number of shrimps sold at the end of the week i. **Economic model (cost of feed consumed per cubic meter during the week *C_i_*):** $C_{i}=p_{F}\;*\;F_{i}$, where $p_{F}$ is the price ($/lb) of feed *and* $F_{i}$ *is the quantity of feed fed per cubic meter* during the week. **Economic model (cost of post-larvae):** $C_{PL}=P_{PL}\;*\;H_{0}$, where $C_{PL}$ is the cost of post-larvae and $H_{0}$ is the quantity purchased at the beginning of the production period. **Economic model (cumulative profit):** $T_{i}=\sum \left(R_{i}-C_{i}\right)-C_{PL}$, where $T_{i}$ is the cumulative profit from the beginning of the batch through the end of week i.	Producer selling price and survival rate can affect the value of net revenue but do not impact the optimal harvesting week. Utility: Determination of the optimal production strategy to maximize the net revenue for a super-intensive recirculating shrimp production system.	[94]
Economic analysis of farm management adjustments as a response to disease risks. A semi-intensive system with different stocking densities of postlarvae/m^2^. **Economic model (Total cost):** Tc=∑px1+px2+px3…pxn, where p is the price and x is the input. **Economic model (profit):** π=gross revenue−total cost **Economic model (breaking yield):** $BEY=\frac{total\;cost}{unitary\;output\;value}$ **Economic model (operating profit margin ratio):** $OPMR=\frac{protfit}{gross\;revenue}\;*\;100$	Upon evaluating different culture time periods (12–31 weeks), results showed that extending culture time periods beyond 25 weeks resulted in lower survival levels, higher feed conversion ratios, and lower profits and operating profit margin ratios. Utility: A bioeconomic model for farm management adjustments as a response to disease risks.	[95]
Prediction of white spot disease effects on the shrimp biomass production through a dynamic stock model. An intensive system with a stocking density of 50–55 postlarvae/m^2^. **Economic model (biomass):** $b_{t}=w_{t}\;*\;n_{t}$**,** where $w_{t}$ is the weight of shrimp and $n_{t}$ is the number of surviving shrimps. **Biological model (weight):** $w_{t}=w_{i}+(w_{f}-w_{i}){\left[\frac{\left(1-k^{t}\right)}{\left(1-k^{c}\right)}\right]}^{3}$, where $w_{i}$ is the initial weight, $w_{f}$ is the final weight, and k relates to the rate at which $w_{t}$ changes from its initial value to its final value. **Biological model (survival):** $n_{t}=n_{0}e^{-Z_{1}t}$**, if** $t\leq t_{w}$ **or** $n_{t}=\left(n_{0}e^{-z_{1}t_{w}}-m\right)e^{-z_{2}\left(t-t_{w}-1\right)}$**, if** $t>t_{w}$, where $n_{t}$ is the number of survivors (%), $n_{0}$ is the initial population, $z_{1}$ is the instantaneous mortality rate previous to the time when die-off from disease occurred ($t_{w}$), m is mortality from disease (%) (hereafter, mortality), and $z_{2}$ is the instantaneous mortality rate after $t_{w}$.	Final weight of shrimp was positively correlated with high pond water temperature and oxygen but inversely correlated with salinity. The dynamics of production predicted with the stock model showed that, for summer, reductions in disease mortalities and delays in the occurrence of mortalities were associated with increasing levels of aeration. Utility: The model can estimate the biomass of shrimp *Litopenaeus vannamei* affected by white spot disease.	[96]
A bioeconomic model to analyze risk and determine whether higher prices for shrimp following the hurricane season would compensate for the risk of hurricane impact. A highly intensive system with a stocking density of 950–1500 (nursery) and 200–580 (growth-out) shrimp/m^2^. **Biological model (growth):** *Same as Ruiz-Velazco* et al. *[84]*. **Survival model:** $n_{t}=n_{0}-z\;*\;t$, where $n_{t}$ is the surviving shrimp at time t, $n_{0}$ is the initial population, and z is the mortality rate. **Technological model (Feed conversion ratio model):** $fcr_{t}=fcr_{i}exp(-mt)$, where $fcr_{t}$ is the feed conversion ratio at time t, $fcr_{i}$ is the initial $fcr_{t}$, and m is the decreasing rate of $fcr_{t}$. **Profit (net revenue) model:** $nr_{t}=i_{t}-c_{t}$, where income ($i_{t}$) is the product of shrimp biomass (b_t_) from the stocking model and shrimp market price and costs ($c_{t}$) are the total costs of postlarvae, feed, labor, harvesting, and miscellaneous costs.	Annual variability of shrimp price, mortality during the second phase of cultivation, and hurricane hazard were the most important risk factors. For a cultivation cycle with a single harvest, net revenue and benefit–cost ratio were maximized by using 1350 postlarvae m^−2^ (nursery) and 300 juveniles m^−2^ and 60 days (grow-out). For a cycle with partial-and-final harvests, maximization was obtained using 1500 postlarvae m^−2^ (nursery), 580 juveniles m^−2^ (phase 1 grow-out), 335 shrimp m^−2^ (phase 2 grow-out). Utility: The bioeconomic model gives recommendations to improve net revenue expectations.	[97]
Determination of profit maximization through the development of a bioeconomic model to analyze the combination of stocking density and date as decision variables for a shrimp farm. A semi-intensive system with a stocking density of 6–30 postlarvae/m^2^. **Biological model (weight increase):** $\frac{dw}{dt}=\Gamma \;*\;f_{1}\left(w\right)\;*\;f_{2}\left(\theta \right)\;*\;f_{3}(r_{n})$, where $f_{1}\left(w\right)$ represents the weight function that incorporates the magnitude of growth as dependent on shrimp weight, $f_{2}\left(\theta \right)$ is the thermal function, which integrates the effect of water temperature and $f_{3}(r_{n})$ is the ration function, which estimates the effect of the feed ration level on shrimp growth. Γ represents the other factors influencing growth and is estimated by calibration. **Biological model (survival):** $N_{\left(t\right)}=N_{0}\;*\;e^{-mt}$, where $N_{0}$ indicates the initial population (t = 0), decreasing exponentially at a constant weekly mortality rate m, and t is the time. **Economic model (profit):** $\pi \left(t_{h}\right)=N\left(t_{h}\right)\;*\;P\left(t_{h}\right)\;*\;w\left(t_{h}\right)-\sum_{t=0}^{t_{h}}(FC(t)+LC(t)+FCv_{we}\left(t\right);FC_{a}\left(t\right))-PLC-FXC+FC_{rp}$, where $N\left(t_{h}\right)$ is the current population; $P\left(t_{h}\right)$ is the selling price; $w\left(t_{h}\right)$ is the shrimp’s weight; PLC is the cost of postlarvae; FC(t) is the cost of feeding; LC(t) is the cost of labor; $FCv_{we}\left(t\right)$, $FC_{a}\left(t\right),$ and $FC_{rp}$ are the cost of fuel; FXC are the fixed costs; and $t_{h}$ is the harvesting time of the culture, which is established by simulation according to the maximum profit condition.	Profit maximization for a semi-intensive shrimp farm was obtained through the development of a bioeconomic model to analyze the combination of stocking density (range: 6–30 PLm^−2^) and date (from March 1st to June 1st) as decision variables for a shrimp farm. The results show that pond water temperatures prevailing during culture cycle when the stocking date is June 1st (temperature in 19-week culture period: 30.76 ± 0.87 °C) and the stocking density is 20–24 PL m^−2^ produce a maximized Present Value Profit (PVπ) of USD^-^ha 10 350 and PVπ USD ha^−1^ 2526 for weekly mortality rates at low (2.1%) and medium (5.8%) levels, respectively. Utility: The optimal selection of stocking density and date in semi-intensive culture of *Litopenaeus vannamei*.	[98]
Estimation of the optimum pond size for intensive commercial production of shrimp through a stochastic bio-economic model. Stocking density of 60–90 postlarvae/m^2^. **Biological model (growth):** $w_{n}=w_{i}+(w_{f}-w_{i}){\left[\frac{\left(1-k^{n}\right)}{\left(1-k^{h}\right)}\right]}^{3}$, where $w_{n}$ is the shrimp weight after n time events have passed, $w_{i}$ is the initial weight, $w_{f}$ is the final weight, k is the growth coefficient, and h is time events that have passed until harvesting time. **Biological model (Survival):** $n_{t}=n_{0}e^{-zt}$, where $n_{t}$ is the number of surviving shrimps at time t, $n_{0}$ is the initial population, z is the mortality rate, and t is the time. **Technological model (feed conversion ratio):** $FCR_{t}=a_{F}t+b_{F}$, where $FCR_{t}$ is the feed conversion ratio at time t and $a_{F}$ and $b_{F}$ are the regression coefficients. **Technological model (Aeration):** $AT_{t}=A_{0}+\frac{A_{F}-A_{0}}{1+e^{d(b-t)}}$, where $AT_{t}$ is the total aeration at time t; $A_{0}$ is the initial aeration, $A_{F}$ is the final aeration, d and b are regression coefficients.	A stochastic bioeconomic model was used to define the optimum pond size for intensive commercial production of *L. vannamei*. Ponds of 2 ha compared to ponds of 8 ha maximized Net Present Value (NPV), Internal Rate of Return (IRR), and Return per Unit Risk (RUR). A total 2 ha pond size with a 10% interest rate showed an NPV of USD 63,300, whereas the 8 ha pond size with an interest rate of 30% showed an NPV of USD 59,800. Utility: Specific recommendation of using 2 ha ponds because of their better economic performance due to larger shrimp size and greater biomass at harvest.	[99]
Analysis of zootechnical, water quality, and management factors influencing intensive production of shrimp when incorporating partial harvesting strategies by deterministic and stochastic models. Stocking density of 30 postlarvae/m^2^. **Biological model (Biomass):** *Same as Ruiz-Velazco* et al. *[84]*. **Biological model (growth):** *Same as Ruiz-Velazco* et al. *[84]*. **Biological (survival):** *Same as Gonzalez Romero* et al. *[87]/Ruiz-Velazco* et al. *[84]*.	The main factors affecting shrimp production were determined by a bioeconomic analysis/modeling as final weight, growth rate, water temperature, pond size, aeration, and time at first partial harvest. Utility: Available models to determine the optimum harvesting strategy.	[100]
Developing a profit maximization model, as an enhancement to the Hanson-Posadas bioeconomic model, to determine the optimal harvesting week, size, and batch number per year and evaluate the economic viability for the super-intensive, biosecure, recirculating shrimp production systems. **Economic model (revenue and cumulative profit):** *same as Zhou [82]* **Biological model (survival):** *same as Zhou [82]* **Growth model:** $w_{i}=\beta_{0}+\beta_{1}t+\beta_{2}t^{2}$, where t is the number of weeks for the production cycle and $\beta_{0}$, $\beta_{1}$, and $\beta_{2}$ are the parameter coefficients.	The outcomes of this study indicated that the optimal parameters for harvesting were within the 12th to 18th week, targeting a size range of 18–21 g and considering three to four batches annually. Utility: Specific economic factors were identified and recommended to improve profitability.	[101]
Determination of the optimal harvesting time that maximizes the net benefits of shrimp cultured in freshwater using a bioeconomic model fitted to different stocking density strategies. An intensive system with stocking densities ranging from 90 to 330 postlarvae/m^2^. **Biological model (Growth):** $\dot{x}(t)={g(x;D}_{0})=a_{0}e^{-a_{1}{{(ln(D}_{0}))}^{2}}x^{2∕3}-a_{2}x$, where $\dot{x}$ is the instantaneous growth rate (g/d), and $a_{0}$, $a_{1}$, and $a_{2}$ are parameters and ln ($D_{0}$) the natural logarithm of the stocking density $D_{0}$. **Biological model (Homogeneous size population):** $n(t)=n_{0}e^{-\mu (D_{0})t}$, where $n_{0}$ is the number of initial individuals at time t = 0 and $D_{0}$ is the stocking density. **Heterogeneous size model (Population model):** **Biological model (Heterogeneous size population):**$N_{t}{\left(t,x\right)+(g(x;D}_{0})N(t,x{))}_{x}=-\mu \left(D_{0}\right)N\left(t,x\right),0<x<\omega ,t>t_{0},N\left(0,x\right)=N_{0}v_{0}\left(x\right),N\left(t,0\right)=0$, where $N_{t}(t,x)$ is the number of individuals dependent on time t and size x, ω>0 is the maximum weight that can be reached by the organism, and $\mu \left(D_{0}\right)$ is the instantaneous mortality rate. In addition, it assumes that throughout the duration of the culture, there is no reproduction or replacement of individuals. $N_{0}$ indicates the initial number of individuals, which are distributed across a range of sizes determined by a beta probability function (was fitted with the data); $v_{0}\left(x\right)$: $v_{O\left(x\right)=}\frac{1}{x_{1}-x_{0}}\;*\;\frac{\Gamma \left(a+\beta \right)}{\Gamma \left(a\right)\Gamma \left(\beta \right)}*{\left(\frac{x-x_{0}}{x_{1}-x_{0}}\right)}^{a-1}*{\left(1-\frac{x-x_{0}}{x_{1}-x_{0}}\right)}^{\beta -1}$, with $\Gamma \left(\alpha \right)=\int_{o}^{\infty }e^{-x}x^{\alpha -1}\mathrm{dx}$ and $x_{0}<x<x_{1}$ and α,β>0. The variable x represents size, Γ is the gamma function, $x_{0}$ and $x_{1}$ represent the upper and lower limits of the initial dispensation growth interval of the individuals, a and β are parameters. **Economic model (Present value of income for homogeneous size and heterogenous size population, respectively):** **=**${V\left(t\right)=e}^{-it}p\left(x\right)x\left(t\right)n(t)$ and $V\left(T\right)=e-it\int_{0}^{\omega }p\left(x\right)xN\left(t,x\right)dx$, where e is the base of the natural logarithmic, t is time, i is the daily discount, n and N are the surviving shrimps (for homogeneous and heterogenous size, respectively), and $p\left(x\right)$ is the market price of the product of size x.	A model was constructed that included the size heterogeneity of the culture and the results compared with the traditional model which assumes size homogeneity for all individuals. The results from both models indicated that the stocking density of 90 shrimp m^−2^ was the best management strategy for optimizing net benefits. Utility: A bioeconomic model and the specific recommendation of considering size heterogeneity in the culture of the white shrimp in freshwater.	[102]
Bioeconomic analysis of super-intensive closed shrimp farming: a case of study in Japan. **Biological model (growth):** Von-Bertalanffy growth model	The population dynamics were limited by unidentified factors differing in water temperature, salinity, dissolved oxygen, and nitrogenous waste. Utility: Practical recommendations for an efficient economic management.	[86]
Determination of the economic feasibility of using probiotics in larval shrimp production and the optimal concentration to maximize the economic performance based on the laboratory production data. A stocking density of 180 larvae/L. **Biological model (survival):** $N_{f}=N_{0}\left(1-eM_{\left(x\right)}^{\prime }\right)$ where $N_{f}$ refers to the number of post-larval whiteleg shrimp at the end of the productive cycle; $N_{0}$ represents the initial number of larvae m^−3^ and $M_{\left(x\right)}^{\prime }$ is an abbreviation of the natural logarithm of mortality observed at the end of the cycle (M) based on the number of probiotics used (x). **Technological model (energy consumption model):** $E_{x}=Eh_{x}+Ew_{x}+E_{a_{x}}$, where $E_{x}$ is the total energy consumed during production; subscript x represents probiotic density; $Eh_{x}$ is the energy needed to increase water temperature; $Ew_{x}$ accounts for the energy needed for water exchange; and $E_{a_{x}}$ is the energy needed to adequately aerate the system. **Economic model (marginal costs):** $MC_{x}=\frac{c_{x+1}-c_{x}}{Nf_{x+1}-Nf_{x}}$, where x is the inoculum added to the medium in g/m^3^ day^−1^ and $c_{x}$ represents the total production costs.	The integration of probiotics into the system yielded an enhancement in survival rates, effectively reducing unit production costs by 44%. However, this improvement was coupled with a modest 6% increase in total production costs. Consequently, the ultimate profitability of this strategy becomes contingent upon the laboratory’s capacity to uphold such an investment. Utility: The analysis revealed that profitability of probiotic strategy depends on the capability of the laboratory to sustain the investment.	[25]
Bioeconomic analysis of the investment in shrimp production in greenhouses using the biofloc system. A highly intensive system with a stocking density of 400 postlarvae/m^2^. **Economic model (net present value): ** $NPV=\sum_{t=1}^{T}\frac{R_{t}-D_{t}}{{\left(1+i\right)}^{t}}+\frac{S_{t}}{{\left(1+i\right)}^{t}}-I_{0}$, where R_t_ is the revenue for a period t; $D_{t}$ is the expense for a period t; $S_{t}$ is the project residual value in the last period; $I_{0}$ is the initial investment; i is the minimum attractiveness rate of return (MARR). **Economic model (Equivalent annual value): ** $EAV=\frac{NPV*1\cdot }{\left[1-{\left(1+i\right)}^{\left(-n\right)}\right]}$, where NPV is the net present value; i is the MARR; n is the number of periods. **Economic model (minimum attractiveness):** $MARR=Rf_{g}+R_{C}+{\beta C}_{LG}[\beta_{GG}+\left(R_{MG}-Rf_{g}\right)](1-R^{2})+InfBR+InfUSA$, where $Rf_{g}$ is the risk-free rate; $R_{C}$ is the country risk; ${\beta C}_{LG}$ is the country beta; $\beta_{GG}$ is the unlevered beta of comparable investments in the market; $R_{MG}$ is the global market return; $R^{2}$ is the coefficient of determination; InfBR is the inflation rate in Brazil; InfUSA is the inflation rate in the USA.	The findings unveiled substantial metrics, including a net present value of USD 904,947.21, a net future value reaching USD 2,401,094.35, an equivalent annual value of USD 148,861.38, a swift payback period of 2 years and 4 months, a discounted payback period of 2 years and 10 months, a robust profitability index of 2.59, a commendable internal rate of return at 41.23%, and a modified internal rate of return of 21.25%. Utility: The analysis demonstrated that *Penaeus vannamei* production using the biofloc technology system is economically sustainable.	[92]

## Data Availability

Data sharing is not applicable to this article as no new data were created or analyzed in this study.

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
