# Peer review of "Use of Immunostimulants in Shrimp Farming—A Bioeconomic Perspective"

_animals, 2025, doi:10.3390/ani15020124_

Round 1
Reviewer 1 Report
Comments and Suggestions for Authors
The topic is quite interesting and the MS is well-written in general. Blow are few minor points:
L53: delete "food"
L329: give the name of the reference as you did in L342.
Author Response
Dear Reviewer:
Thank you for your valuable comments and for allowing us to revise and improve this manuscript. We have now addressed all suggestions in this revised version, and we hope this version will now be acceptable for publication. Detailed responses to all suggestions are provided below. All changes have been highlighted in red in the manuscript for easy spotting.
The topic is quite interesting and the MS is well-written in general.
Reply: Thank you.
Blow are few minor points:
L53: delete "food"
Reply: Done.
L329: give the name of the reference as you did in L342.
Reply: Done. It was included in the new version “Moreno-Figueroa et al. [87]”.

Reviewer 2 Report
Comments and Suggestions for Authors
In this paper, the authors provide some methods to choice the proper immunostimulants in shrimp farming. The work is necessary and significant. However, here are some points you need to answer.
1.Please provide information on the different diseases of shrimp farming and the resulting losses.
2.The authors discussed the cost of immunostimulants, please provide information on the names, prices, functions, and recommended dosages of well-established immunostimulants on the market in the section 3. Immunostimulants for shrimp aquaculture.
3.The information in Table 1 is not concise enough, please make changes to make it clear to readers.
4.Lines 316-318.“This section may be divided by subheadings. It should provide a concise and precise description of the experimental results, their interpretation, as well as the experimental conclusions that can be drawn.”It should be removed?
5.Funding: This research was funded by CONAHCYT, grant number 565382, to H.R.N-A.).
Is this bracket redundant?
Comments on the Quality of English LanguageThe language of this article needs to be improved.
Author Response
Dear Reviewer:
Thank you for your valuable comments and for allowing us to revise and improve this manuscript. We have now addressed all suggestions in this revised version, and we hope this version will now be acceptable for publication. Detailed responses to all suggestions are provided below. All changes have been highlighted in red in the manuscript for easy spotting.
In this paper, the authors provide some methods to choice the proper immunostimulants in shrimp farming. The work is necessary and significant. However, here are some points you need to answer.
Reply: Thank you.
1.Please provide information on the different diseases of shrimp farming and the resulting losses.
Reply: We provided the following information in the introduction section:
“In the Aquatic Animal Health Code (WOAH 2024), the priority listed diseases for shrimp are Acute hepatopancreatic necrosis disease, decapod iridescent virus 1 disease, Enterocytozoon hepatopenaeicausing disease, Infectious hypodermal and haematopoietic necrosis disease, Infectious myonecrosis virus disease, Necrotising hepatopancreatitis (Hepatobacter Penaei), Taura syndrome virus disease, White spot disease, and Yellow head disease. In this regard, the estimated annual revenue loss in India due to Enterocytozoon hepatopenaei and white spot syndrome virus diseases was US$ 567.62 and US$ 238.33 M, respectively (Patil et al., 2021), which can vary when analyzed at the farm level (Geetha et al., 2022). Similarly, net revenue losses due to acute hepatopancreatic necrosis disease ranged from US$ -727.56–672.48 ha-1 (Estrada-Perez et al., 2020). In a study, the economic impact of infectious myonecrosis virus disease accounted for 29.86% mortality and US$ -24822.76 ha-1 (Kusna et al., 2023). Another study estimated that stocking low-level Infectious hypodermal and haematopoietic necrosis virus in shrimp augmented the farm gate value (US $67,000 ha-1) with respect to the higher level one (Sellars et al., 2019). Estimated losses decades ago for Taura syndrome virus and Yellow head diseases have been around US$ 0.5-2.0 billion (Lightner et al., 2012). In general, the shrimp industry encounters high losses in aquaculture due to diseases. Diseases have created economic crises, making sustainable production difficult and obligating geographic relocation in the absence of cost-effective preventive and curative treatments (Asche et al., 2021).”
Lightner, D. V., Redman, R. M., Pantoja, C. R., Tang, K. F. J., Noble, B. L., Schofield, P., ... & Navarro, S. A. (2012). Historic emergence, impact and current status of shrimp pathogens in the Americas. Journal of invertebrate pathology, 110(2), 174-183.
Sellars, M. J., Cowley, J. A., Musson, D., Rao, M., Menzies, M. L., Coman, G. J., & Murphy, B. S. (2019). Reduced growth performance of Black Tiger shrimp (Penaeus monodon) infected with infectious hypodermal and hematopoietic necrosis virus. Aquaculture, 499, 160-166.
Kusna, M., Prayitno, S. B., & Wijayanto, D. (2023). Economic impact due to infectious myonecrosis virus (IMNV) disease in intensive vannamei shrimp aquaculture in Kendal Regency. Aquaculture, Aquarium, Conservation & Legislation, 16(5), 2637-2647.
Asche, F., Anderson, J. L., Botta, R., Kumar, G., Abrahamsen, E. B., Nguyen, L. T., & Valderrama, D. (2021). The economics of shrimp disease. Journal of invertebrate pathology, 186, 107397.
Geetha, R., Avunje, S., Solanki, H. G., Priyadharshini, R., Vinoth, S., Anand, P. R., ... & Patil, P. K. (2022). Farm-level economic cost of Enterocytozoon hepatopenaei (EHP) to Indian Penaeus vannamei shrimp farming. Aquaculture, 548, 737685.
Estrada-Perez, N., Ruiz-Velazco, J. M., & Hernández-Llamas, A. (2020). Economic risk scenarios for semi-intensive production of Litopenaeus (Penaeus) vannamei shrimp affected by acute hepatopancreatic necrosis disease. Aquaculture Reports, 18, 100442.
Patil, P. K., Geetha, R., Ravisankar, T., Avunje, S., Solanki, H. G., Abraham, T. J., ... & Vijayan, K. K. (2021). Economic loss due to diseases in Indian shrimp farming with special reference to Enterocytozoon hepatopenaei (EHP) and white spot syndrome virus (WSSV). Aquaculture, 533, 736231.
World Organisation for Animal Health - Aquatic Animal Health Code. (2024) https://www.woah.org/en/what-we-do/standards/codes-and-manuals/aquatic-code-online-access/?id=169&L=1&htmfile=index.htm
2.The authors discussed the cost of immunostimulants, please provide information on the names, prices, functions, and recommended dosages of well-established immunostimulants on the market in the section 3. Immunostimulants for shrimp aquaculture.
Reply: It is an interesting observation. We did a search of commercial immunostimulants for shrimp aquaculture, and we found that most are probiotics, followed by polysaccharides and few of them are herbal medicines that are combined with other products. Examples are provided in this new version as you recommended.
3.The information in Table 1 is not concise enough, please make changes to make it clear to readers.
Reply: Done. The Table 1 length was reduced to half of the original, being more concise and clearer for readers.
4.Lines 316-318.“This section may be divided by subheadings. It should provide a concise and precise description of the experimental results, their interpretation, as well as the experimental conclusions that can be drawn. ”It should be removed?
Reply: You are right. We removed.
5.Funding: This research was funded by CONAHCYT, grant number 565382, to H.R.N-A.).
Is this bracket redundant?
Reply: Yes, you are right. We removed.

Reviewer 3 Report
Comments and Suggestions for Authors
line 171: please provide examples of plants and isolates if possible.
line 316-318
line 339: could the enhancement not also be due to temperature-elevated metabolic rate?
p. 427-439: this simply a description of a controlled experiment (can be deleted).
Table 1 is a great summary of models "on the shelf." Is there any way of evaluating the utility of some or all in prediction?
Author Response
Dear Reviewer:
Thank you for your valuable comments and for allowing us to revise and improve this manuscript. We have now addressed all suggestions in this revised version, and we hope this version will now be acceptable for publication. Detailed responses to all suggestions are provided below. All changes have been highlighted in red in the manuscript for easy spotting.
Line 171: please provide examples of plants and isolates if possible.
Reply: As you recommended, examples were included in the section 3.2:
In addition, plants including species from families such as Meliaceae, Quillajaceae, Fabaceae, Lauraceae, Araliaceae, Apiaceae, Campanulaceae, Asteraceae, Rosaceae, Asparagaceae, Liliaceae, and Myrtaceaehave been used as natural immunostimulants [48]. In this context, polysaccharides derived from plant fruits [49] and root extracts [50, 51] alone or mixed with drugs have been evaluated in shrimps and revealed immunostimulant activity and enhanced disease resistance [52].
We used the following supporting references:
- Filho, Luiz G. A., Dos Santos.; Diniz,F,M.; Pereira, Alitiene M.L. Chapter 9 - Immunostimulants derived from plants and algae to increase resistance of pacific white shrimp (Litopenaeus vannamei) against vibriosis. Editor: Atta-ur-Rahman, Studies in Natural Products Chemistry 2023, 77 ,pp. 297-337, ISSN 1572-5995, ISBN 9780323912945, https://doi.org/10.1016/B978-0-323-91294-5.00009-9
- Pholdaeng, K.; Pongsamart, S. Studies on the immunomodulatory effect of polysaccharide gel extracted from Durio zibethinus in Penaeus monodon shrimp against Vibrio harveyi and WSSV. Fish Shellfish Immunol2010, 28(4), pp. 555-561.https://doi.org/10.1016/j.fsi.2009.12.009
- Pan, S.; Jiang, L.; Wu, S. Stimulating effects of polysaccharide from Angelica sinensis on the nonspecific immunity of white shrimps (Litopenaeus vannamei). Fish Shellfish Immunol 2018, 74, pp. 170-174. https://doi.org/10.1016/j.fsi.2017.12.067
- Liu, XL.; Xi, QY.; Yang, L.; et al. The effect of dietary Panax ginseng polysaccharide extract on the immune responses in white shrimp, Litopenaeus vannamei. Fish Shellfish Immunol 2011, 30(2), pp.495-500. https://doi.org/10.1016/j.fsi.2010.11.018
- Zhai, Q.; Li, J.; Feng, Y.; Ge, Q. Evaluation of combination effects of Astragalus polysaccharides and florfenicol against acute hepatopancreatic necrosis disease-causing strain of Vibrio parahaemolyticus in Litopenaeus vannamei. Fish Shellfish Immunol 2019, 86, pp. 374-383. https://doi.org/10.1016/j.fsi.2018.11.065
line 316-318
Reply: You are right. We removed this part.
line 339: could the enhancement not also be due to temperature-elevated metabolic rate?
Reply: Yes, you are right. We eliminated this sentence to avoid confusion. We focused on the results of this study, and an appropriate sentence was incorporated into the new version.
“Overall, productivity and economic performance accounting for the net revenue was mainly influenced by the shrimp price, followed by the final weight of shrimp, dissolved oxygen, and temperature.”
- 427-439: this simply a description of a controlled experiment (can be deleted).
Reply: We agree. We removed this part.
Table 1 is a great summary of models "on the shelf." Is there any way of evaluating the utility of some or all in prediction?
Reply: In the “Highlights” column of Table 1, we have incorporated each cited study's utility according to your recommendation.

Round 2
Reviewer 2 Report
Comments and Suggestions for Authors
The authors answered all the questions I asked, so I agree with the article acceptance.